# Current Status of the Self-Expandable Metal Stent as a Bridge to Surgery Versus Emergency Surgery in Colorectal Cancer: Results from an Updated Systematic Review and Meta-Analysis of the Literature

**DOI:** 10.3390/medicina57030268

**Published:** 2021-03-15

**Authors:** Roberto Cirocchi, Alberto Arezzo, Paolo Sapienza, Daniele Crocetti, Davide Cavaliere, Leonardo Solaini, Giorgio Ercolani, Antonio V. Sterpetti, Andrea Mingoli, Enrico Fiori

**Affiliations:** 1Department of Medicine and Surgery, University of Perugia, 05100 Terni, Italy; roberto.cirocchi@unipg.it; 2Department of Surgery, Turin University, 10133 Torino, Italy; alberto.arezzo@unito.it; 3Department of Surgery, University of Rome, 00161 Rome, Italy; daniele.crocetti@uniroma1.it (D.C.); antonio.sterpetti@uniroma1.it (A.V.S.); andrea.mingoli@uniroma1.it (A.M.); enrico.fiori@uniroma1.it (E.F.); 4General and Oncologic Surgery, Morgagni-Pierantoni Hospital, AUSL Romagna, 47121 Forlì, Italy; d.cavaliere@ausl.fo.it; 5Department of Medical and Surgical Sciences (DIMEC), University of Bologna, 40126 Bologna, Italy; leonardosolaini@gmail.com (L.S.); giorgio.ercolani2@unibo.it (G.E.)

**Keywords:** self-expandable metal stent, colonic obstructions, emergency surgery

## Abstract

*Background*: The current use of endoscopic stenting as a bridge to surgery is not always accepted in standard clinical practice to treat neoplastic colonic obstructions. *Objectives:* The role of colonic self-expandable metal stent (SEMS) positioning as a bridge to resective surgery versus emergency surgery (ES) for malignant obstruction, using all new data and available variables, was studied and we focused on short- and long-term results. *Materials and Methods:* A systematic review with meta-analysis was performed. PubMed, SCOPUS and Web of Science databases were included. The search comprised only randomized controlled trials (RCTs) investigating the interventions that included SEMS positioning versus ES. The primary outcomes were the rates of overall postoperative mortality, clinical and technical success. The secondary outcomes were the short- and long-term results. *Results:* A total of 12 studies were eligible for further analyses. A laparoscopic colectomy was the most common operation performed in the SEMS group, whereas the traditional open approach was commonly used in the ES group. Intraoperative colonic lavage was seldomly performed during ES. There were no differences in mortality rates between the two groups (RR 1.06, 95% CI 0.55 to 2.04; I^2^ = 0%). In the SEMS group, the rate of successful primary anastomosis was significantly higher in of SEMS (69.75%) than in the ES (55.07%) (RR 1.26, 95% 245 CI 1.01 to 1.57; I^2^ = 86%). Conversely, the upfront Hartmann procedure was performed more frequently in the ES (39.1%) as compared to the SEMS group (23.4%) (RR 0.61, 95% CI 0.45 to 0.85; I^2^ = 23%). The overall postoperative complications rate was significantly lower in the SEMS group (32.74%) than in the ES group (48.25%) (RR 0.61, 95% CI 0.41 to 0.91; I^2^ = 65%). *Conclusions*: In the presence of malignant colorectal obstruction, SEMS is safe and associated with the same mortality and significantly lower morbidity than the ES group. The rate of successful primary anastomosis was significantly higher than the ES group. Nevertheless, recurrence and survival outcomes are not significantly different between the two groups. The analysis of short- and long-term results can suggest the use of SEMS as a bridge to resective surgery when it is performed by an endoscopist with adequate expertise in both colonoscopy and fluoroscopic techniques and who performed commonly colonic stenting.

## 1. Introduction

While the self-expandable metal stent (SEMS) is commonly accepted in a palliative setting for obstructive colorectal cancer, deciding whether to proceed with endoscopic stent as a bridge to curative surgery or upfront emergency surgery (ES) in case of symptomatic left-sided malignant colonic obstruction is still under debate. Several authors [1,2] do not recommend the use of SEMS before surgery in resectable patients because it may harm long-term outcomes. Similarly, the 2017 Guidelines of the World Society of Emergency Surgery [3] recognize “interesting advantages” offered by the use of the SEMS, but they highlighted that its use for surgically treatable cases may expose some long-term oncologic issues. Conversely, the recent European Society of Gastrointestinal Endoscopy (ESGE) Guideline [4] recommended the use of SEMS because it is associated with lower mortality rate, shorter hospital stay and a lower rate of related colostomy.

To date, only a few randomized controlled trials (RCTs) have been published on this topic. The last systematic review including only RCTs was published three years ago [5], and the authors did not report a pooled analysis on the survival variables. Furthermore, an additional RCT was published by Elwan et al. [6] in 2020 adding data for future analysis. Recently, the long-term oncologic results of the ESCO Trial were presented [7].

We aimed to perform a systematic review on SEMS as a bridge to surgery versus ES for malignant left-sided colonic obstruction, using all new data and available variables and focusing our analysis on short- and long-term results.

## 2. Materials and Methods

This systematic review and meta-analysis were performed following the Preferred Reporting Items for Systematic Reviews and Meta-Analyses (PRISMA) guidelines [8].

### 2.1. Criteria to Satisfy for a Study to Be Included in the Meta-Analysis:

Inclusion criteria: RCTsTypes of participants: patients with intestinal obstruction due to colon or rectal cancer.Types of treatment: SEMS as a bridge to surgery versus ES.Exclusion criteria: endoscopic or surgical treatment performed only for palliation.

### 2.2. The Following Primary Outcomes Were Observed:

The postoperative overall mortality ratePostoperative complications rateSuccessful of primary anastomosis

### 2.3. The Secondary Outcomes Were as It Follows:

#### 2.3.1. Short-Term Outcomes:

Technical success rate (avoidance of colonic perforation, bleeding or stent migration in SEMS group and intraoperative surgical complications in ES group)The clinical success rate (intended as colonic decompression)Anastomotic leakage rateUpfront Hartmann procedure or another derivative colostomy ratePermanent Hartmann procedure or another derivative colostomy rate

#### 2.3.2. Long-Term Outcomes:

Overall recurrenceLocal recurrenceSystemic recurrence3-years overall survival (OS)3-years disease free survival (DFS)

The literature search was performed 21 November 2020 on the following databases: PubMed, SCOPUS and Web of Science (WOS) to identify all eligible studies. The combination of the following words was used: “large bowel obstruction” or “colonic obstruction)”, “colorectal stent” or “colon stent” or “rectal stent”. No language restrictions were applied.

The selected studies’ title and abstracts were independently screened by two authors (E.F. and R.C.); successively full-text articles of potentially relevant studies were evaluated independently by the same two authors (E.F. and R.C.).

When overlapping was found between multiple articles published by the same authors and no difference in the examined time, only the most recent trial was enclosed to avoid duplication. The PubMed function “related articles” and Google Scholar database were used to find further articles. A search on Google book was performed for the analysis of the grey literature (https://books.google.com accessed on 21 November 2020).

Two authors (E.F. and R.C.) developed a data extraction sheet based on the model of the “Cochrane Consumers and Communication Review Group” and independently extracted data from the included studies [9].

The assessment of methodological quality was performed independently by two authors (R.C. and E.F.), who assessed the methodological quality of the included studies using the methods described in the Cochrane Handbook for Systematic Reviews of Interventions for Randomized Controlled Trials (RCTs) [10].

### 2.4. Statistical Analysis

Data were analyzed for risk ratios (RR) in the case of dichotomous variables, and in weighted mean differences (WMD) for continuous variables. Intention-to-treat analysis was performed. The randomized Mantel–Haenszel method was used for the meta-analysis [11]. All results were displayed in a forest plot graph. The I2 test was utilized for the heterogeneity assessment. A value greater than 50% was significant for heterogeneity. The data analysis was performed using the meta-analysis software Review* Manager (RevMan) v5.4.1 (Copenhagen: The Nordic Cochrane Centre, The Cochrane Collaboration, 2020).

## 3. Results

The PRISMA flow chart for systematic review schematically reported (Figure 1). Briefly, after this screening for relevance, 20 articles remained for further assessment of eligibility. Eight of them were successively excluded [12,13,14,15,16,17,18,19] and a total of 12 articles were eligible for further analyses (Table 1) [6,7,20,21,22,23,24,25,26,27,28,29]. Noticeably, 3 RTCs reported the short-term results in the first publication [20,23,27] and the long-term outcomes in a subsequent publication [7,28,29]. Therefore, for long-term outcomes we considered the second publication reporting an up-to-date follow-up.

### 3.1. Characteristics of the Studies Included

The majority of studies were performed in Europe (4 studies: 299 patients, 54.36%), followed by Asia (3 studies: 131 patients, 23.82%) and Africa (2 study: 120 patients, 21.82%).

Three studies were multicentric while the remaining were single center. All articles describe the duration of the participants’ enrollment comprised between 3 and 12 years. The studies were published between 2009 and 2020.

Three RCTs were prematurely terminated for the unacceptable high complication rate; the first was terminated because emergency surgery group had significantly increased rate of anastomotic leak [21]; the second reported a significantly higher incidence of 30-day morbidity in the SEMS group of patients [23]; the third for a high rate of colonic perforations during stent placement and a high rate of technical failure of stent placement [24]. Patients’ characteristics were similar between the groups (Appendix A). Four studies included stage IV patients and in two, the inclusion rate was different (Appendix A) [20,25]. The tumor location was reported in all but one [20] study; in seven the cancer was located in the left colon or rectum, and in one [6] the authors also included patients affected with right colon cancer (Appendix A).

The laparoscopic colectomy was commonly performed in the SEMS group; conversely, in ES group a traditional open approach was preferred (S3). The intraoperative colonic lavage was performed in a few studies during ES (Appendix A).

In ES group, surgical treatment varied deeply; total or subtotal abdominal colectomy with ileorectal anastomosis, Hartmann’s procedure, colorectal resection with primary anastomosis, or derivative colostomy (Appendix A). Conversely, all patients undergoing SEMS positioning as a bridge to resective surgery had colorectal resection with primary anastomosis.

In eight studies the type of stent was reported, and in most of the studies the authors used the Wallflex stents; the time intercourse between stent placement and elective surgery was 5 to 10 days (Appendix A). The perforation rate varied between 8.9–14% (Appendix A).

### 3.2. Quality Assessment of the Included Studies

The potential risk for bias in each of the trials and a summary of these using the criteria and the “Risk of bias” table are reported in Cochrane Handbook for Systematic Reviews of Interventions Version 5 [9,10]. The risk of bias of RCTs was reported in Appendix A. (review authors’ judgments about each risk of bias item presented as percentages across all included studies) and Appendix A (review authors’ judgments about each risk of bias item for each included study).

### 3.3. Primary Outcomes

#### 3.3.1. Overall Postoperative Mortality Rate

Eight studies including 508 patients (252 SEMS and 256 ES) reported the mortality rates. We registered 16 (6.35%) deaths in the SEMS group and 17 (6.64%) in the ES group. Four studies did not specify when mortality occurred [6,20,22,25]. Three studies reported an overall in-hospital mortality rate [21,24,26], one study a 30-day mortality [23] and one a 60-day mortality [27]. No differences between the mortality rate in the two groups (RR 1.06, 95% CI 0.55 to 2.04; I^2^ = 0%) (Figure 2a) were recorded.

The subgroup analysis of hospital mortality [21,24,26] reported the same mortality in the two groups (RR 1.23, 95% CI 0.13 to 11.32; I^2^ = 30%) (Figure 2b)

#### 3.3.2. Postoperative Complications Rate

Nine studies (567 patients: 281 SEMS and 286 ES) reported the postoperative complications. The overall postoperative complications rate was significantly lower in the SEMS group (32.74%) and in the ES group (48.25%) (RR 0.61, 95% CI 0.41 to 0.91; I^2^ = 65%) (Figure 3a). Two studies did not specify when the complications occurred [6,25]. Two studies [21,26] reported an overall in-hospital postoperative complication rates without further specification, one study [23] a 30-day and one [27] a 60-day postoperative complication rate.

The subgroup analysis of hospital postoperative complications [21,26] showed a statistically significant lower complication rate in the SEMS group (RR 0.26, 95% CI 0.12 to 0.58; I^2^ = 0%) as compared to the ES group (Figure 3b).

#### 3.3.3. Clinical Success Rate. Successful Primary Anastomosis

Nine studies (557 patients: 281 SEMS and 276 ES) reported this outcome. The rate of primary anastomosis was significantly higher in of SEMS (69.75%) than in the ES (55.07%) (RR 1.26, 95% CI 1.01 to 1.57; I^2^ = 86%) (Figure 4).

### 3.4. Secondary Outcomes

#### 3.4.1. Short-Term Outcomes

##### Technical Success Rate

Eight studies (508 patients: 252 SEMS and 256 ES) reported the clinical success rate. The technical success in SEMS group was intended as absence of colonic perforation, bleeding or stent migration, whereas in the ES group was intended as the absence of intraoperative surgical complications. The failure rate in SEMS group was 10.7%, whereas in the ES group there were no intraoperative surgical complications (RR 12.05, 95% CI 2.83 to 51.23; I^2^ = 0%).

##### Clinical Success Rate

Eight studies including 508 patients (252 SEMS and 256 ES) reported the clinical success rate intended as colonic decompression. The colonic decompression was significantly higher in patients who underwent ES (100%) than in of the patients undergoing SEMS (86.5%) (RR 9.18, 95% CI 3.06 to 27.59; I^2^ = 0%).

##### Anastomotic Leakage Rate

Eight studies (345 patients: 190 SEMS and 155 ES) reported the anastomotic leakage rate. It was lower in SEMS group (5.8%) as compared to the ES group (7.7%) (RR 0.78, 95% CI 0.32 to 1.91; I^2^ = 4%) (Figure 5).

##### Upfront Hartmann Procedure or Another Derivative Colostomy Rate

Eight studies (508 patients: 252 SEMS and 256 ES) reported the Hartmann or derivative colostomy procedure rate. The Hartmann or derivative colostomy procedure rate was statistically higher in ES group (39.1%) when compared to the SEMS group (23.41%) (RR 0.62, 95% CI 0.45 to 0.85; I^2^ = 23%) (Figure 6).

##### Permanent Hartmann Procedure or Another Derivative Colostomy Rate

Five studies (324 patients: 164 SEMS and 160 ES) reported the covering stoma. The covering stoma rate was higher in the ES group (35.62%) as compared to the SEMS group (22.56%) (RR 0.64, 95% CI 0.33 to 1.25; I^2^ = 47%) (Figure 7).

#### 3.4.2. Long-Term Outcomes

##### Overall Recurrence

The overall recurrence rate was reported in five studies (302 patients: 148 SEMS and 154 ES). The rate was higher in the ES group (24.67%) as compared to the SEMS group (35.14%) (RR 1.63, 95% CI 0.88 to 3.04; I^2^ = 57%) (Figure 8).

##### Local Recurrence

The local recurrence rate was reported in three studies (225 patients: 108 SEMS and 117 ES). The recurrence rate was higher in the SEMS group (11.11%) as compared to the ES group (8.54%) (RR 1.34, 95% CI 0.52 to 3.43; I^2^ = 0%) (Figure 9).

##### Systemic Recurrence

The systemic recurrence rate was reported in three studies (225 patients: 108 SEMS and 117 ES). The rate was not statistically significant different between the two groups (21.77% in SEMS group vs 20% in ES group) (RR 1.01, 95% CI 0.60 to 1.71; I^2^ = 0%) (Figure 10).

##### Three Years OS

The 3-year OS was reported in four studies (235 patients: 118 SEMS and 117 ES). Three years’ survival rate was higher in ES group (73.5%) when compared to SEMS group (67.8%), but the result was not statistically significant (RR 1.20, 95% CI 0.80 to 1.79; I^2^ = 33%) (Figure 11).

##### Three Years DFS

The 3-year DFS was reported in three studies (204 patients: 102 SEMS and 102 ES). The DFS rate was better in the ES group (63.46%) when compared to the SEMS group (58.65%) (RR 1.22, 95% CI 0.87 to 1.69; I^2^ = 0%) (Figure 12).

## 4. Discussion

The use of SEMS as a bridge to curative surgery is still controversial because it has some advantages but also some disadvantages [30]. Our up-to-date systematic review and meta-analysis demonstrated that the use of SEMS is associated with low in-hospital mortality, high rate of primary anastomosis and decreased need for Hartmann procedure or derivative colostomies.

Data regarding mortality greatly vary among the different published systematic reviews [31,32,33,34,35,36,37]. However, the most recent meta-analyses on this topic seem to be in line with our results, demonstrating a benefit in terms of mortality rates with the use of SEMS when compared to ES [38,39]. Particularly, our results on mortality rates were obtained from RCTs, which conferred a robust evidence in favor of SEMS in the presence of malignant left-sided colonic obstruction as a bridge to surgery.

The RCTs included in the papers are somewhat different. In effect, the recent systematic review and meta-analysis performed from Spannenburg et al. ES [38] included studies (RCTs and CCTs) in which the surgical treatment was performed with curative or palliative aim. Differently, our review included only RCTs in which the patients underwent curative surgery.

The higher rate of primary anastomosis in the SEMS group as compared to the ES group is also a clear advantage of this treatment, and our observations are consistent with the majority of previous studies (RR 1.26, 95% CI 1.01 to 1.57; I^2^ = 86%). Analyzing the available literature, we also should take into account when choosing one of the two different strategies: SEMS or ES both the clinical success rate (defined as the ability of the procedure to decompress the bowel) and the technical success rate (defined as intraprocedural versus intraoperative complications). This point represents an important limitation of this meta-analysis for the few different definitions of outcomes such as technical and clinical success in the included studies. For this reason, we have aggregated similar conditions reported in the literature in two outcome groups of this review (Appendix A).

Our analysis suggests a preferential use of ES when considering these variables. Furthermore, SEMS might be a more complex and challenging procedure that is operator-dependent, affected by the expertise in operative endoscopy and it should be reserved to a tertiary care center.

Overall, our results support the use of SEMS whenever feasible, leaving the choice of ES for patients at a high risk of clinical/technical failure. At present, few studies tried to investigate the predictors of technical failure, but it seems that a stenosis greater than 8 cm in length and the need for endoscopic guidance may be associated with higher rates of technical and/or clinical stenting failure [40]. Finally, we believe that further analyses are required in order to identify and select the patients who might benefit from SEMS prior to resective surgery.

The main disadvantage in deploying SEMS as a bridge to surgery is the possibility to jeopardize long-term outcome [2,39,41,42,43,44]. Some authors sustain the hypothesis that SEMS deployment may cause microperforation leading to a higher risk of peritoneal carcinomatosis [39,41,42,43,44]; others support the possibility that a tumor’s compression by SEMS causes tumoral spreading into the nearby vessels, favoring hematogenous diffusion. However, if these hypotheses have had a reliable basis, our study should have produced consistent evidence in decreased survival and disease-free survival in the SEMS group.

Moreover, some authors [7,45] reported a higher rate of harvested lymph nodes, although not reaching statistical significance, in the resected patients after SEMS deployment, assuming that there was a strict relationship between delayed surgery and the availability of a more experienced colorectal surgeon in an elective setting.

However, our analysis found that the overall recurrence along with local and systemic recurrence and the three-year overall survival rate were similar among the two groups. According to these observations, we are confident to reinforce the use of SEMS in the presence of malignant left-sided colonic obstruction.

The quality of life, a crucial variable [46,47,48,49] which at least theoretically might favor the SEMS group of patients, was not considered in the RCTs included in the present analysis. Therefore, further studies are warranted to investigate the impact of the stoma creation rate, the risk of reintervention and the incidence of persistent stomas on patient-reported outcome.

## 5. Conclusions

In the presence of malignant colorectal obstruction, SEMS is safe and associated with the same mortality and significantly lower morbidity than the ES group. The rate of successful primary anastomosis was significantly higher than the ES group. Nevertheless, recurrence and survival outcomes are not significantly different between the two groups. The analysis of short- and long-term results can suggest the use of SEMS as a bridge to resective surgery when it is performed by an endoscopist with adequate expertise in both colonoscopy and fluoroscopic techniques and who performed commonly colonic stenting.

The role of the operator’s experience for optimal success in stent placement (operator-dependent) and the need for specific training should be emphasized, as well as the importance of experienced colorectal surgeons’ readiness in elective compared to emergency surgery.

## Figures and Tables

**Figure 1 medicina-57-00268-f001:**
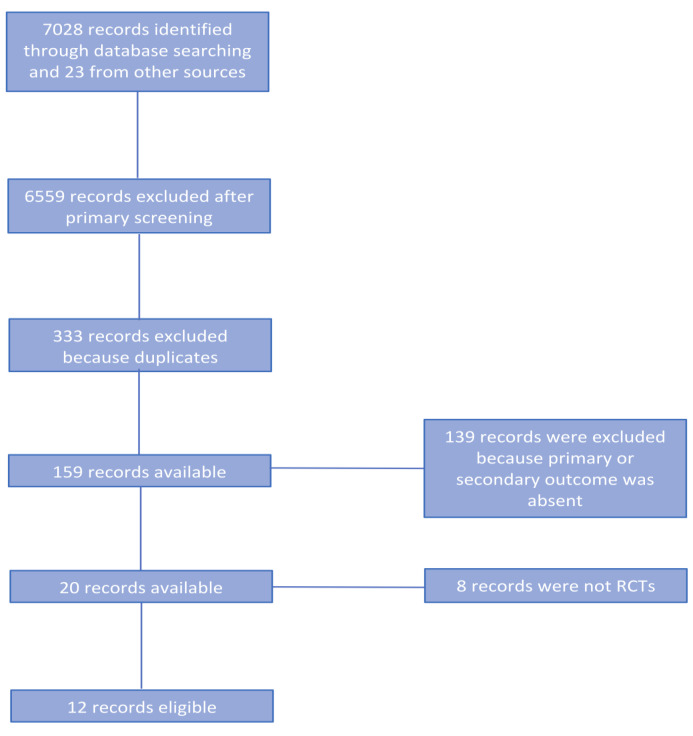
PRISMA flow chart.

**Figure 2 medicina-57-00268-f002:**
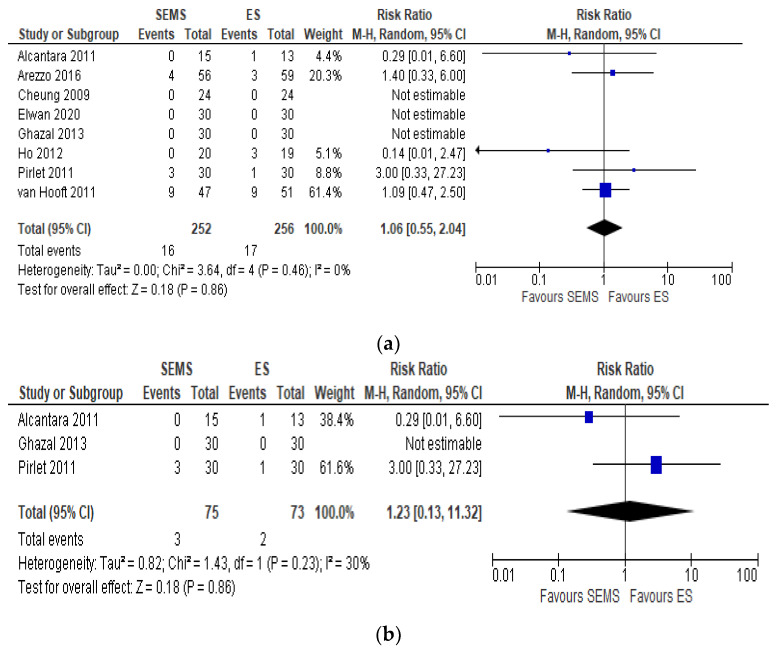
(**a**) Forest plot of overall postoperative mortality rate. (**b**) Forest plot of overall postoperative mortality during the hospital stay.

**Figure 3 medicina-57-00268-f003:**
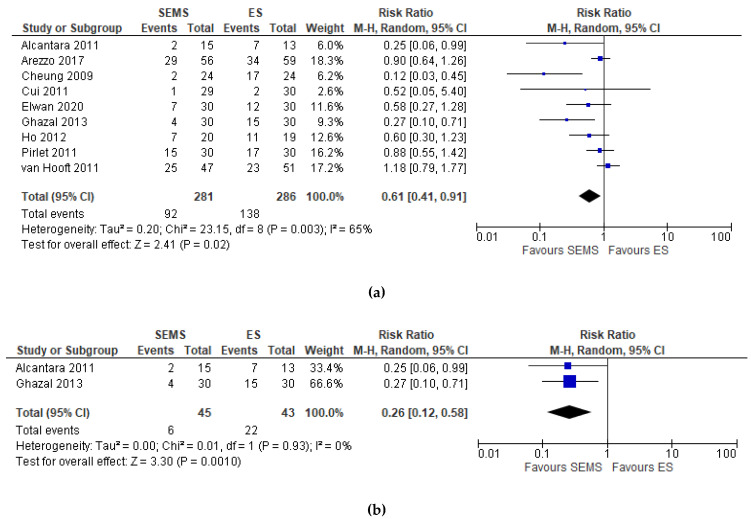
(**a**) Forest plot of overall postoperative complications. (**b**) Forest plot of overall postoperative complications during hospital stay.

**Figure 4 medicina-57-00268-f004:**
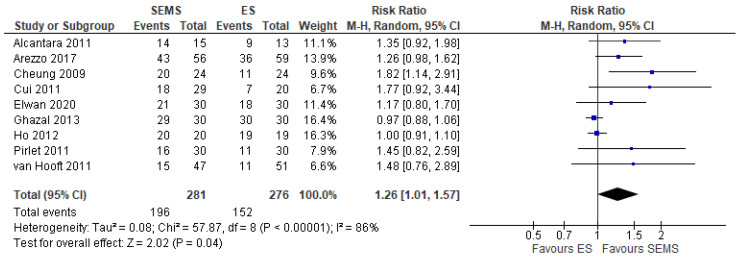
Forest plot of success of primary anastomosis.

**Figure 5 medicina-57-00268-f005:**
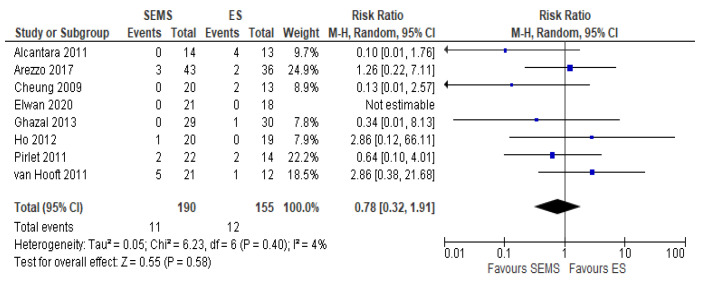
Forest plot of anastomotic leak.

**Figure 6 medicina-57-00268-f006:**
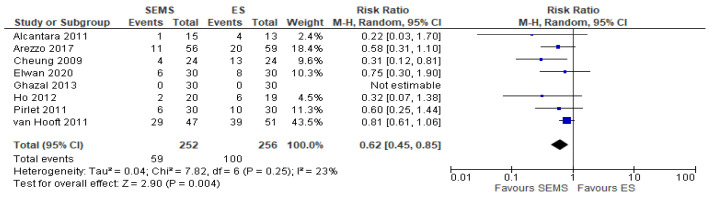
Forest plot of upfront Hartmann procedure or another derivative colostomy rate.

**Figure 7 medicina-57-00268-f007:**
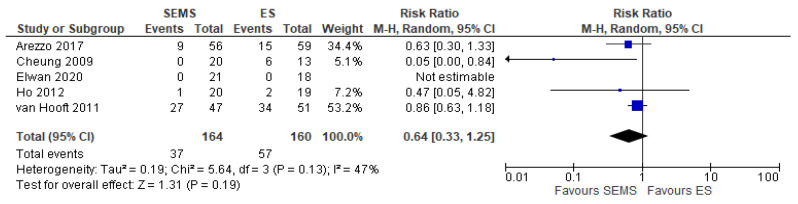
Forest plot of permanent Hartmann procedure or another derivative colostomy rate.

**Figure 8 medicina-57-00268-f008:**
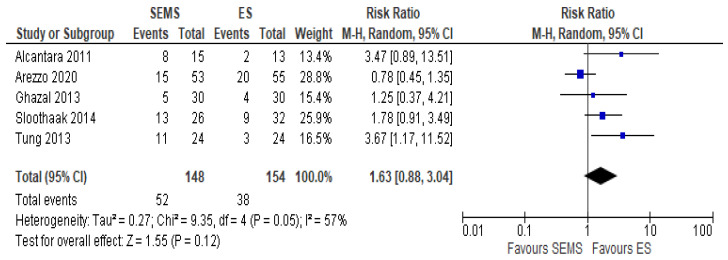
Forest plot of overall recurrence rate.

**Figure 9 medicina-57-00268-f009:**
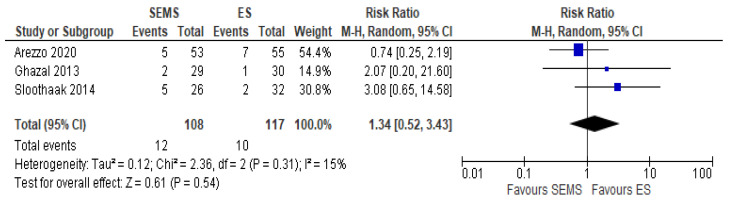
Forest plot of local recurrence rate.

**Figure 10 medicina-57-00268-f010:**
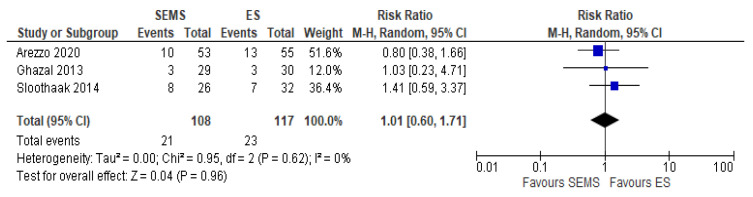
Forest plot of systemic recurrence rate.

**Figure 11 medicina-57-00268-f011:**
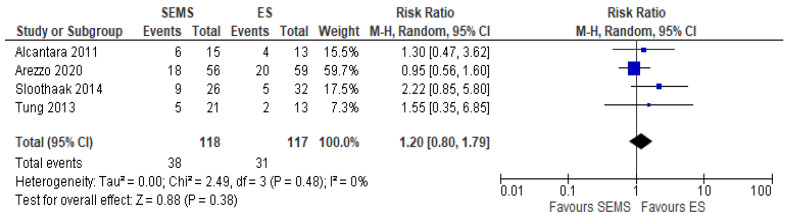
Forest plot of overall survival.

**Figure 12 medicina-57-00268-f012:**
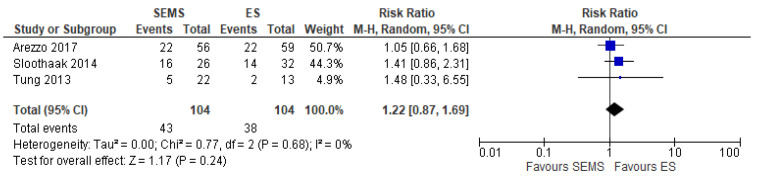
Forest plot of 3-year Disease Free Survival.

**Table 1 medicina-57-00268-t001:** Characteristics of included studies.

Author	Country	Number of Centres	Time of Enrollment	Premature Closure of the Trial	Number of Patients Enrolled	
					SEMS	Surgery
Arezzo et al., 2020	Italy/Spain	Multicenter	2008–2015	No	56 *	59
Elwan et al., 2020	Egypt	Single-centre	2015–2019	No	30	30
Arezzo et al., 2017	Italy/Spain	Multicenter	2008–2015	No	56 *	59
Sloothaak et al., 2014	Netherlands	Single-centre	2007–2009	Yes	26	32
Thung et al., 2013	Hong Kong, China	Single-centre	2002–2005	No	24	24
Ghazal et al., 2013	Egypt	Single center	2009–2012	No	30	30
Ho et al., 2012	Singapore	Single-centre	2004–2008	No	20	19
Pirlet et al., 2011	France	Multicenter	2002–2006	Yes	30	30
Van Hooft et al., 2011	Netherlands	Multicenter	2007–2009	Yes	47	51
Cui et al., 2011	China	Single center	2005–2009	No	29	15
Alcántara et al., 2011	Spain	Single-centre	2004–2006	Yes	15	13
Cheung et al., 2009	Hong Kong, China	Single-centre	2002–2005	No	24	24

* Two patients underwent self-expandable metal stent (SEMS) positioning but refused resective surgery.

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
