# Peer review of "Current Status of the Self-Expandable Metal Stent as a Bridge to Surgery Versus Emergency Surgery in Colorectal Cancer: Results from an Updated Systematic Review and Meta-Analysis of the Literature"

_medicina, 2021, doi:10.3390/medicina57030268_

Round 1
Reviewer 1 Report
This is a meta-analysis of SEMS vs. emergency surgery for colorectal cancer. The topic is important for treatment selection and is also discussed in the ESGE guidelines.
- There is another meta-analysis in the same topic (European Journal of Surgical Oncology 46 (2020) 1404-1414). RCTs included in the paper are somewhat different. Any differences in the inclusion criteria.
- Difficulties in the meta-analysis lie in the different definition of outcomes such as technical and clinical success among studies. Did the authors encounter those issues? If so, please discuss.
- This meta-analysis is focused on the comparison between SEMS and emergency surgery. The title should be changed.
Author Response
On behalf of all the co-authors, we want to thank you for reviewing our manuscript titled Current status of self-expandable metal stent as bridge to surgery for colorectal cancer: results from an updated systematic review and meta-analysis of literature, manuscript ID: medicina-1097287.
The manuscript has been revised according to the reviewers’ comments.
Reviewer #1.
Comments:
This is a meta-analysis of SEMS vs. emergency surgery for colorectal cancer. The topic is important for treatment selection and is also discussed in the ESGE guidelines.
- There is another meta-analysis in the same topic (European Journal of Surgical Oncology 46 (2020) 1404-1414). RCTs included in the paper are somewhat different. Any differences in the inclusion criteria.
Responses: The manuscript from Spannenburg et coll. included studies (RCTs and CCTs) in which the surgical treatment was performed with curative or palliative aim. Differently our review included only RCTs in which the patients underwent to curative surgery.
Thanks for your advices to improve the discussion with the review of Spannenburg. We have revised the discussion section following the suggestions.
We added in the discussion: “The RCTs included in the papers are somewhat different. In effect the recent systematic review and meta-analysis performed from Spannenburg et coll. ES [38] included studies (RCTs and CCTs) in which the surgical treatment was performed with curative or palliative aim. Differently our review included only RCTs in which the patients underwent to curative surgery”.
Comments:
- Difficulties in the meta-analysis lie in the different definition of outcomes such as technical and clinical success among studies. Did the authors encounter those issues? If so, please discuss.
We performed a new table (Supplement 6) and added in the discussion: “This point represents an important limitation of this the meta-analysis for the few different definitions of these two outcomes such as technical and clinical success among studies. For this reason, we have aggregated similar conditions reported in the literature in two outcome’ groups of this review (Supplement 6)”
|
|
Aggregate definitions of clinical success |
Aggregate definitions of technical success |
|
Cheung 2009 |
“Resolution of obstruction” |
“Appropriate placement of the stent” |
|
Alcantara 2011 |
“The success of the procedure was defined as the clinical appearance of intestinal transit and the disappearance of the obstruction on abdominal radiography” |
“…complications arose with stent placement…”- |
|
Van Hooft 2011 |
“…obstruction clinically resolved….” |
“Major stent-procedure and stent-related complications were perforation, stent migration and reobstruction” |
|
Pirlet 2011 |
“Clinical failure, defined as a lack of bowel decompression within the first 3 postprocedure days” |
“The reason for the technical failure was inability to pass the stricture with the guidewire, malfunction of the stent delivery system and observation of direct colonic perforation” |
|
Ho 2012 |
“Clinical success was defined as the colonic decompression within 96 h after successful placement of the stent, with passage of stools and resolution of nausea and vomiting, and confirmed on plain abdominal radiograph” |
“Technical success was defined as successful SEMS placement and deployment” |
|
Ghazal 2013 |
“In the ESER group, patients had upfront endoscopic placement, under fluoroscopic guidance, of a colonic stent across the obstruction according to the standard technique described elsewhere. Following successful stent placement, the patient was admitted to a general surgical ward, received a colonic purge, and subsequently underwent elective tumor resection and primary anastomosis within 7–10 days of stent placement” |
“……failure of passage of the guide-wire through the obstructing lumen…..”
|
|
Arezzo 2017 |
“Clinical success was defined as resolution of occlusive symptoms by gas and faeces passage” |
“Technical success was defined as correct stent placement under radiographic and endoscopic vision” |
|
Elwan 2020 |
“Successful decompression was defined by the improvement of obstructive manifestations as patients passing flatus or stools and/or disappearance of nausea and vomiting, and no air-fluid levels on plain abdominal radiograph” |
Not reported a definition or data of the outcome |
Comments:
- This meta-analysis is focused on the comparison between SEMS and emergency surgery. The title should be changed.
Thanks for your comments on the title of manuscript. We have revised the title as following:
Current status of the self-expandable metal stent as a bridge to surgery versus emergency surgery in colorectal cancer: results from an updated systematic review and meta-analysis of the literature.

Reviewer 2 Report
Thank you for the opportunity to review the paper titled “Current status of the self-expandable metal stent as a bridge to surgery for colorectal cancer: results from an updated systematic review and meta-analysis of the literature”
The Analysis is performed well, the statistics are adequate, but I have some major comments as follows:
#1 In the abstract, the description in conclusion does not match the contents of results. Authors should change the description to be consistent.
#2 The references No. 3 and 4 cited in the introduction section should be replaced with new ones, respectively.
No.3→van Hooft JE, et al. Self-expandable metal stents for obstructing colonic and extracolonic cancer: European Society of Gastrointestinal Endoscopy (ESGE) guideline – update 2020. Endos 2020;52(5):389–407.
No.4→Pisano et al. 2017 WSES guidelines on colon and rectal cancer emergencies: obstruction and perforation. World J Emerg Surg 2018;13:36
In updated Guideline of European Society of Gastrointestinal Endoscopy, BTS using colonic stent is recommended. This is due to the improved outcomes of colonic stenting, gradually. Whereas, in the 2017 Guidelines of the World Society of Emergency Surgery do not explicitly recommend BTS by colonic stent for obstructive left-sided colorectal cancer.
Authors should revise the introduction section and the discussion section.
#3 Last paragraph of the Discussion: As for patient’s QOL, I think it would be easier to understand if authors describe it specifically, including stoma creation rate, the risk of re-intervention and the incidence of persistent stomas.
Author Response
On behalf of all the co-authors, we want to thank you for reviewing our manuscript titled Current status of self-expandable metal stent as bridge to surgery for colorectal cancer: results from an updated systematic review and meta-analysis of literature, manuscript ID: medicina-1097287.
Reviewer #2.
The Analysis is performed well, the statistics are adequate, but I have some major comments as follows:
- #1 In the abstract, the description in conclusion does not match the contents of results. Authors should change the description to be consistent.
Thanks for your comments on the title of manuscript. We have revised the title as following: “In the presence of malignant colo-rectal obstruction, SEMS is safe and associated with the same mortality and primary anastomosis rate than the ES group. Nevertheless, Recurrence and survival outcomes are not significantly different between the two groups. The analysis of short- and long-term results cannot suggest the use of SEMS as a bridge to resective surgery when it is possible.”
- #2 The references No. 3 and 4 cited in the introduction section should be replaced with new ones, respectively.
No.3→van Hooft JE, et al. Self-expandable metal stents for obstructing colonic and extracolonic cancer: European Society of Gastrointestinal Endoscopy (ESGE) guideline – update 2020. Endos 2020;52(5):389–407.
No.4→Pisano et al. 2017 WSES guidelines on colon and rectal cancer emergencies: obstruction and perforation. World J Emerg Surg 2018;13:36
We have changed the references suggested.
- In updated Guideline of European Society of Gastrointestinal Endoscopy, BTS using colonic stent is recommended. This is due to the improved outcomes of colonic stenting, gradually. Whereas, in the 2017 Guidelines of the World Society of Emergency Surgery do not explicitly recommend BTS by colonic stent for obstructive left-sided colorectal cancer.
Authors should revise the introduction section and the discussion section.
Authors’ reply: A revision of the text according to the updated guidelines was performed and added in the introduction section and discussion section.
- #3 Last paragraph of the Discussion: As for patient’s QOL, I think it would be easier to understand if authors describe it specifically, including stoma creation rate, the risk of re-intervention and the incidence of persistent stomas.
Authors’ reply: A clearer descriptions was provided, as suggested.

Round 2
Reviewer 2 Report
I have reviewed the revised manuscript.
The revised manuscript is well described I have no hesitation in recommending it for publication.